# A missense variant in *SLC39A8* is associated with severe idiopathic scoliosis

Gabe Haller[1], Kevin McCall[1], Supak Jenkitkasemwong[2], Brooke Sadler[3], Lilian Antunes[1], Momchil Nikolov[1], Julia Whittle[1], Zachary Upshaw[1], Jimann Shin[4], Erin Baschal[5], Carlos Cruchaga[6], Matthew Harms[7], Cathleen Raggio[8], Jose A. Morcuende[9], Philip Giampietro[10], Nancy H. Miller[11], Carol Wise [12,13,14], Ryan S. Gray [15], Lila Solnica-Krezel[4], Mitchell Knutson[2], Matthew B. Dobbs[1,16] & Christina A. Gurnett[1,3,17]

Genetic factors predictive of severe adolescent idiopathic scoliosis (AIS) are largely unknown. To identify genetic variation associated with severe AIS, we performed an exome-wide association study of 457 severe AIS cases and 987 controls. We find a missense SNP in *SLC39A8* (p.Ala391Thr, rs13107325) associated with severe AIS ($P = 1.60 \times 10^{-7}$, OR = 2.01, CI = 1.54–2.62). This pleiotropic SNP was previously associated with BMI, blood pressure, cholesterol, and blood manganese level. We replicate the association in a second cohort (841 cases and 1095 controls) resulting in a combined $P = 7.02 \times 10^{-14}$, OR = 1.94, CI = 1.63–2.34. Clinically, the minor allele of rs13107325 is associated with greater spinal curvature, decreased height, increased BMI and lower plasma manganese in our AIS cohort. Functional studies demonstrate reduced manganese influx mediated by the *SLC39A8* p. Ala391Thr variant and vertebral abnormalities, impaired growth, and decreased motor activity in *slc39a8* mutant zebrafish. Our results suggest the possibility that scoliosis may be amenable to dietary intervention.

[1] Department of Orthopaedic Surgery, Washington University, St. Louis, MO, USA. [2] Food Science and Human Nutrition Department, University of Florida, Gainesville, FL, USA. [3] Department of Neurology, Washington University, St. Louis, MO, USA. [4] Department of Developmental Biology, Washington University School of Medicine, St. Louis, MO, USA. [5] Department of Orthopaedic Surgery, University of Colorado, Denver, CO, USA. [6] Department of Psychiatry, Washington University, St. Louis, MO, USA. [7] Department of Neurology, Columbia University, New York, NY, USA. [8] Hospital for Special Surgery, New York, NY, USA. [9] Department of Orthopaedics and Rehabilitation, University of Iowa, Iowa City, IA, USA. [10] Department of Pediatrics, Drexel University, Philadelphia, PA, USA. [11] Department of Orthopaedic Surgery, University of Colorado, Denver, CO, USA. [12] Sarah M. and Charles E. Seay Center for Musculoskeletal Research, Texas Scottish Rite Hospital for Children, Dallas, TX, USA. [13] Departments of Orthopaedic Surgery, University of Texas Southwestern Medical Center at Dallas, Dallas, TX, USA. [14] McDermott Center for Human Growth and Development, University of Texas Southwestern Medical Center at Dallas, Dallas, TX, USA. [15] Department of Pediatrics, Dell Pediatric Research Institute, Dell Medical School, The University of Texas at Austin, Austin, TX, USA. [16] Shriners Hospital for Children, St. Louis, MO, USA. [17] Department of Pediatrics, Washington University, St. Louis, MO, USA. These authors contributed equally: Matthew B. Dobbs, Christina A. Gurnett. Correspondence and requests for materials should be addressed to C.A.G. (email: gurnettc@neuro.wustl.edu)

Scoliosis is the most common pediatric musculoskeletal disorder, with ~0.3% of children having scoliosis with a spinal curvature of >20° (Cobb angle) requiring treatment[1], and more than 1 in 10,000 children developing severe spine deformity requiring surgery[2]. The majority of scoliosis is considered idiopathic with onset in adolescence, while other cases occur as a consequence of disorders such as Marfan syndrome, cerebral palsy, or muscular dystrophy[3] that have distinguishing features in addition to scoliosis. While the etiology of adolescent idiopathic scoliosis (AIS) is not well understood, recent genome-wide association studies have identified common variants near neural cell adhesion molecules[4], ladybird homeobox 1 (LBX1)[5], G-protein-coupled receptor 126 (GPR126)[6], and the transcription factor BNC2[7] that are modestly associated with AIS. Additionally, we recently demonstrated the collective impact of rare genetic variants in several extracellular matrix genes (FBN1, FBN2, and musculoskeletal collagen genes)[8–10] on AIS risk. However, a vast majority of the heritability of AIS remains unexplained.

Genetic factors responsible for scoliosis curve progression have been hypothesized to be independent from those associated with its initiation[11], yet few genetic risk factors for scoliosis curve progression have been identified, largely due to the rarity of severe scoliosis cases. To date, the only genetic associations identified for severe AIS (Cobb angle > 40°) are with rare coding variants in FBN1 and FBN2 and common variants near SOX9 and KCNJ2[9,12]. A Japanese genome-wide association study identified a common variant in MIR4300HG (the host gene of microRNA MIR4300) that influences scoliosis curve progression[13], but additional studies are needed to develop population-specific prevention strategies that can be applied for short durations during the adolescent growth period when spinal curves progress.

Here we present an exome-wide association study of severe adolescent idiopathic scoliosis and identify an association with a missense SNP in the gene SLC39A8. We replicate this finding in a second independent cohort and show that in addition to scoliosis risk, the SNP is associated with several related traits in our AIS cohort including height, BMI, and curve severity. We further show that the minor allele affects manganese transport in vitro and that disruption of slc39a8 in zebrafish leads to vertebral abnormalities, impaired growth, and decreased motor activity.

## Results

**Genetic association analysis.** To identify common coding variants that confer susceptibility to severe, progressive AIS, we performed exome sequencing on a cohort of 457 unrelated European American AIS cases and 987 unrelated European American controls. The AIS cohort consisted predominantly of individuals with severe, surgical range scoliosis (mean Cobb angle = 52°, range: 20–180). A total of 52,480 single-nucleotide polymorphisms (SNPs) with minor allele frequency (MAF) > 1% passed

quality control criteria (see Methods) and were subjected to single-variant association analysis.

Two variants, both within the SLC39A8 gene, yielded associations with AIS that passed the threshold for exome-wide significance ($P < 9.52 \times 10^{-7}$) (rs13107325 logistic $P = 1.60 \times 10^{-7}$; OR = 2.01, CI = 1.54–2.62). (Fig. 1). These two SNPs (rs13107325 and rs13105581) are in strong linkage disequilibrium within our sample ($r^2 = 0.9$, $D' = 1$). The SNP rs13107325 encodes a non-synonymous variant (p.Ala391Thr) in the SLC39A8 gene that encodes the divalent cation transporter ZIP8 (Supplementary Fig. 1). The other significant SNP, rs13105581, is located within an intron of SLC39A8. We additionally compared the allele frequency in AIS cases to a large control cohort, consisting of individuals of European descent (Non-Finnish) from the Exome Aggregation Consortium (ExAC) ($N = 34,057$) and observed an allele frequency at rs13107325 consistent with that of controls (MAF = 0.067). Additionally, the association of rs13107325 with AIS remained when only severe AIS patients were used in the analysis ($N = 409$) (logistic $P = 7.7 \times 10^{-8}$; OR = 2.08, CI = 1.58–2.73).

To confirm the association between AIS and rs13107325, this SNP was genotyped in an independent cohort of European American subjects with moderate to severe (mean Cobb angle = 44°, range: 20–104) adolescent idiopathic scoliosis (841 AIS cases and 1095 controls). We found strong evidence for association of rs13107325 with severe AIS in this independent cohort (logistic $P = 9.97 \times 10^{-9}$; OR = 2.00, 95% confidence interval (CI) = 1.57–2.57) and in the combined sample ($P = 7.02 \times 10^{-14}$; OR = 1.94, 95% CI = 1.63–2.34) (Table 1).

**Effect of SLC39A8 A391T on clinical characteristics.** To understand the mechanism by which the SLC39A8 p.A391T allele leads to an increased risk of scoliosis, we investigated the clinical characteristics of patients harboring the risk allele. We observed a dose-dependent correlation between SLC39A8 391T allele count and scoliosis curve severity as measured by the Cobb angle (Fig. 2a). The minor allele was also associated with decreased height and increased body-mass index (Fig. 3b, c), consistent with prior work[14]. Because a previous genome-wide association had identified the minor allele of rs13107325 as being associated with reduced blood $Mn^{2+}$ levels[15], we obtained plasma $Mn^{2+}$ and $Fe^{2+}$ levels from a subset of our AIS cohort. Heterozygous carriers of rs13107325 had significantly lower plasma $Mn^{2+}$ levels compared to non-carriers ($P = 0.01$, $t$-test), but there was no difference in plasma $Fe^{2+}$ levels ($P = 0.84$, $t$-test) as expected (Fig. 2). We observed no association between carrier status at rs13107325 and scoliosis family history, age, sex, Beighton joint hypermobility scores, or systemic score from the Ghent criteria[16,17].

**In vitro functional testing.** Despite the fact that rs13107325, and SLC39A8 more broadly, has been implicated in numerous

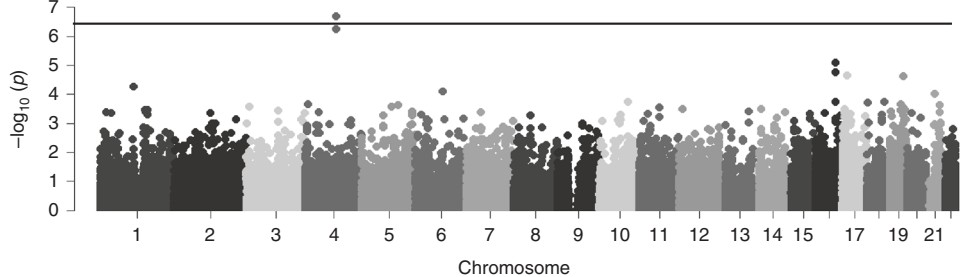

**Fig. 1** Manhattan plot showing AIS exome-wide association results. Linear regression p-values (−log10) for 52,480 single-nucleotide polymorphisms with minor allele frequency > 0.01. One SNP (rs13107325) was significantly associated with AIS after Bonferroni correction for multiple tests

**Table 1 Association of rs13107325 with adolescent idiopathic scoliosis**

| Group | Number of samples | | MAF | | P-value | OR (95% CI) |
|---|---|---|---|---|---|---|
| | Case | Control | Case | Control | | |
| AIS GWAS | 457 | 987 | 0.14 | 0.08 | $1.60 \times 10^{-7}$ | 2.01 (1.54–2.62) |
| AIS replication | 841 | 1095 | 0.11 | 0.06 | $9.97 \times 10^{-9}$ | 2.00 (1.57–2.57) |
| Combined | 1314 | 2082 | 0.12 | 0.07 | $7.02 \times 10^{-14}$ | 1.94 (1.63–2.34) |

MAF = minor allele frequency, OR = odds ratio, CI = confidence interval, AIS = adolescent idiopathic scoliosis

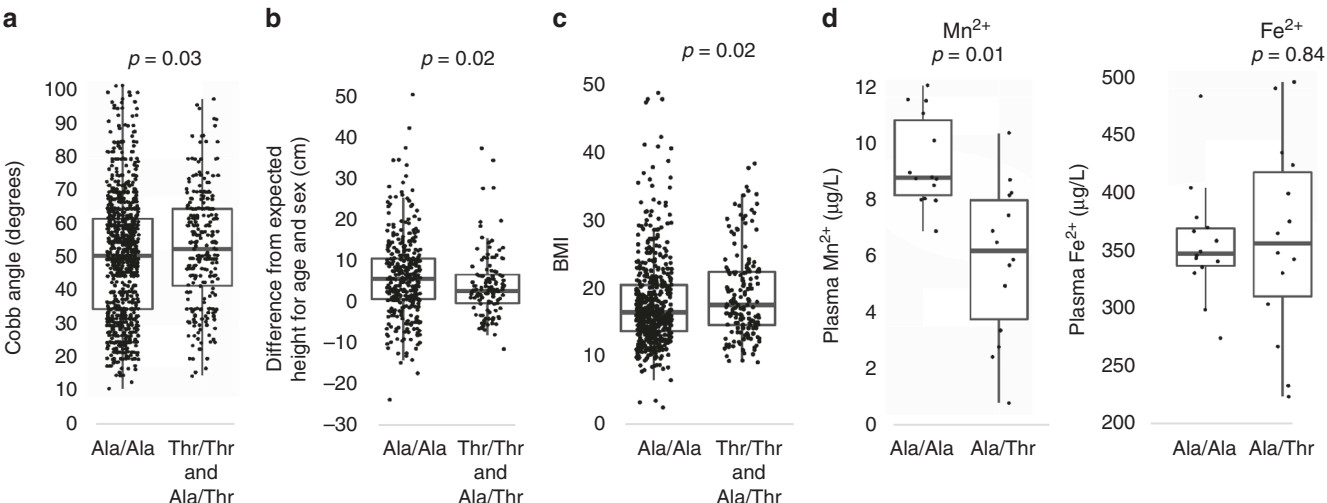

**Fig. 2** Clinical characteristics of AIS cases with different rs13107325 genotypes. Comparison of **a** scoliosis curve severity (Cobb angle), **b** height, **c** body-mass index, and **d** plasma manganese and iron levels in AIS patients who are carriers of the *SLC39A8* 391T allele compared to AIS cases who are non-carriers. For Cobb angle, Ala/Ala ($N = 940$) were compared to the combined Ala/Thr ($N = 242$) and Thr/Thr ($N = 26$). For height, Ala/Ala ($N = 361$) were compared to the combined Ala/Thr and Thr/Thr carriers ($N = 123$). For plasma manganese and iron, only Ala/Ala ($N = 14$) and Ala/Thr ($N = 14$) were measured due to lack of plasma samples from Thr/Thr carriers. Lines are median, 1st and 3rd quartiles. Error bars are 95% confidence intervals. All comparisons were performed using *t*-tests

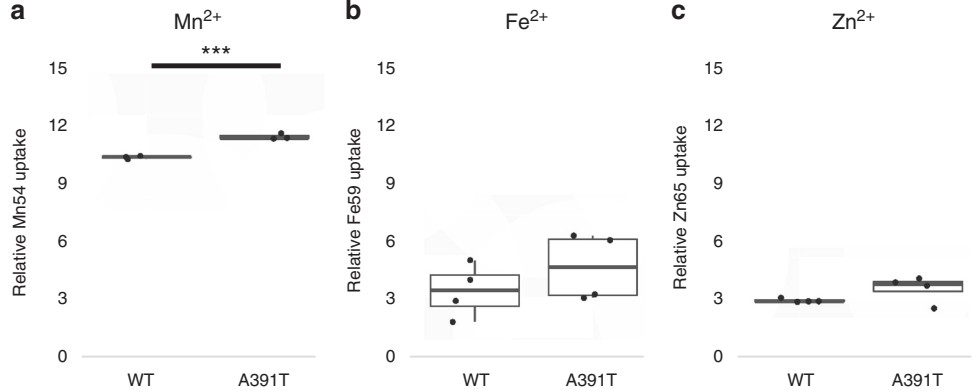

**Fig. 3** Cellular uptake of ions with *SLC39A8* overexpression in HEK293 cells. Comparison of uptake of either radio-labeled **a** Mn$^{2+}$, **b** Fe$^{2+}$, or **c** Zn$^{2+}$ with overexpression of either vector control, *SLC39A8* wild-type or *SLC39A8* A391T in HEK293 cells. All measurements were the average of three replicate experiments. Lines are median, 1st and 3rd quartiles. Error bars are 95% confidence intervals. All comparisons were performed using *t*-tests. ***$P = 5 \times 10^{-4}$

complex traits, little is known about the functional effects of this coding variant on protein function. The minor allele was recently reported to be associated with reduced *SLC39A8* mRNA expression in human liver[18], but the functional SNP within the LD bin has yet to be determined. SLC39A8, also known as ZIP8, is a divalent cation importer capable of transporting Zn$^{2+}$, Fe$^{2+}$, or

Mn$^{2+}$ [19]. To determine the effects of *SLC39A8* p.A391T allele on divalent cation transport, we overexpressed either human *SLC39A8* wild-type or p.A391T in HEK293 cells and measured the ability of cells to import Zn$^{2+}$, Fe$^{2+}$, or Mn$^{2+}$. Overall, cells transfected with *SLC39A8* increased influx of all cations studied. The transporter had the greatest effect on Mn$^{2+}$, with

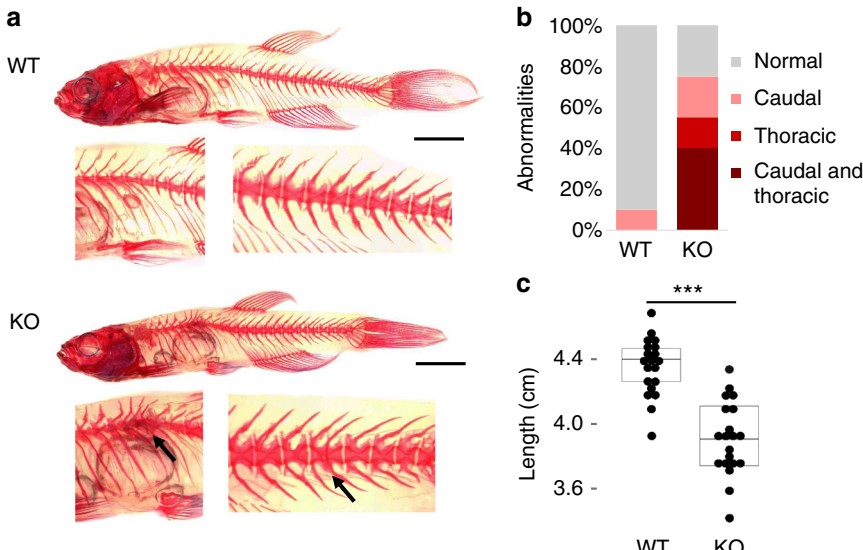

**Fig. 4** Skeletal abnormalities and growth deficit in *slc39a8*−/− zebrafish. **a** Thoracic and caudal vertebral malformations causing mild spinal curvature in *slc39a8*−/− zebrafish (bottom) compared to WT (top) stained with alizarin red. Magnified images show vertebral fusions in KO fish. Scale bar 5 mm. **b** *slc39a8*−/− zebrafish have more spinal abnormalities than WT fish. **c** Reduced length of *slc39a8*−/− zebrafish ($N = 20$) compared to wild-type ($N = 20$) at 9 mpf ***$p < 10^{-6}$, *t*-test. Lines are mean, first and third quartiles

overexpression increasing $Mn^{2+}$ uptake by more than tenfold. Cells expressing the *SLC39A8* p.A391T allele transported less $Mn^{2+}$ than wild-type ($P = 5 \times 10^{-4}$, *t*-test) (Fig. 3). Intracellular transport of both $Zn^{2+}$ and $Fe^{2+}$ was similar between wild-type and p.A391T overexpressing cells, consistent with human data showing predominant effects of rs13107325 on $Mn^{2+}$ blood levels, with little effect on other metal ions[15].

**Effects of *slc39a8* disruption in zebrafish.** Finally, to study the phenotypes associated with loss of *slc39a8* function, we engineered and isolated a mutation predicted to be deleterious via the CRISPR-Cas9 system. Zebrafish harboring a 17 bp tandem duplication (c.1058_1074dup) predicted to lead to premature termination of the protein were bred to homozygosity and used in all further experiments. Among homozygous *slc39a8* mutant fish, we observed a significantly increased proportion of fish at 9 months post-fertilization (mpf) with thoracic or caudal vertebral malformations or both ($P = 3 \times 10^{-5}$, *t*-test) (Fig. 4a, b; Supplementary Fig. 2). Often the thoracic malformations involved vertebral bunching leading to mild spinal curvature. We also observed similar rates of skeletal abnormalities among genotyped homozygous *slc39a8* mutant fish from a heterozygote by heterozygote cross at 12 weeks post-fertilization (Supplementary Fig. 3). These vertebral malformations are similar to those seen in *leviathan*/*col8a1a* mutant fish and embryos treated with lysyl oxidases that develop severe late-onset scoliosis[20]. Unlike mutant *col8a1a* fish, we did not observe any abnormalities in the notochord by calcein staining at 10 dpf, however (Supplementary Fig. 4). We also observed significantly decreased body length ($P = 2 \times 10^{-7}$, *t*-test) (Fig. 4c) among *slc39a8* mutant fish consistent with gene disruption in mice, which resulted in small pups with early postpartum lethality[21]. Further, *slc39a8* mutant fish displayed markedly decreased motor activity and startle responses at 4 days post-fertilization (dpf). Remarkably the movement deficits observed in these mutant larvae were rescued upon 24-h exposure to increased manganese [500 mM], a dose with minimal effect on wild-type larvae (Fig. 5). Additionally, we performed whole-mount in situ hybridization in wild-type 4 dpf zebrafish larvae to determine the expression pattern of *slc39a8* in vivo and observed

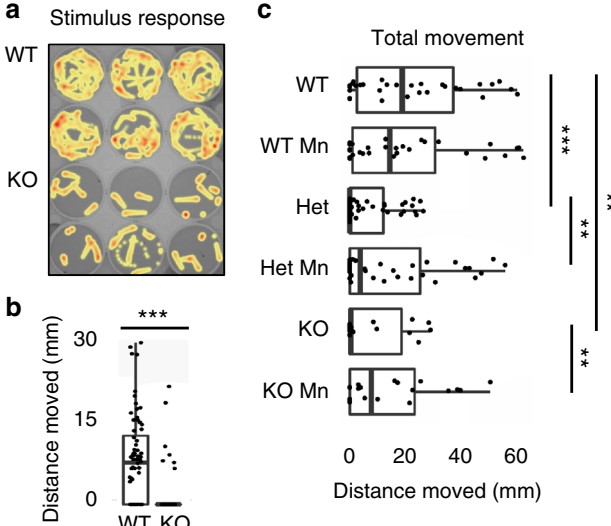

**Fig. 5** Movement deficits in zebrafish *slc39a8*−/− larvae. Zebrafish movements were quantified for 25 min without disturbing (total movement) followed by a tap (stimulus response) at 4 dpf. **a**, **b** Stimulus response visualized as a heat map following tap showing reduced movement in *slc39a8*−/− larvae compared to WT. **c** Total movement is less in *slc39a8* hets and *slc39a8*−/− larvae compared to WT but normalizes with addition of 200 mM Mn for 24 h with little effect on WT. ***$p < 0.001$. **$p < 0.01$, *t*-test. Lines are median, 1st and 3rd quartiles. Error bars are 95% confidence intervals. All comparisons were performed using *t*-tests

strong expression in the eye, brain, spinal cord, and endodermal tissues (Supplementary Fig. 5).

## Discussion

We identified a highly significant association between a coding variant in the manganese transporter *SLC39A8* and risk of

adolescent idiopathic scoliosis. This association was replicated in a second independent cohort of AIS patients and controls. We observe an odds ratio near 2 in both the discovery cohort and replication cohort, which is large for genome-wide association studies. The effect size of the SNP may still be overestimated, however, justifying further study in additional cohorts of AIS patients. SNP rs13107235 was previously implicated as influencing a number of metabolic traits, including blood HDL cholesterol level, blood pressure, body-mass index and most recently schizophrenia and blood manganese levels[15,18,22–25] and is arguably one of the most highly pleiotropic variants in the human genome[26]. In addition to the dose-dependent association of this SNP with degree of spinal curvature, we observed associations between height, BMI and blood manganese levels in our AIS cohort, consistent with previous studies. We do not have these variables measured in controls. In future studies, incorporation of these variables as covariates or performing Mendelian randomization analysis might allow us to determine the causal relationships between these traits, each of which correlate with genotype at rs13107325. Currently, it is unknown whether genotype at this SNP effects scoliosis risk directly or indirectly through its effect on other traits.

Manganese deficiency, while rare in humans, causes growth impairment and skeletal abnormalities in animals[27]. Recessive mutations of SLC39A8 were recently shown to cause cognitive impairment, epilepsy, hearing loss, and skeletal abnormalities that include short limbs and scoliosis[28,29]. Skeletal abnormalities have been partially attributed to the role of manganese as the preferred cofactor for multiple enzymes involved in cartilage and bone development, including glycosyltransferases[30].

Interestingly, the two AIS associated SNPs are monomorphic in Asian or African populations, explaining the lack of association in prior AIS genome-wide association studies, largely in Asian populations[6,7,12,31]. In addition, our cohort is composed mostly of AIS cases with severe scoliosis, possibly making the genetic load in this cohort higher than cohorts used in previous studies. Since our study was restricted to variants within the exome, most of the previously AIS associated common variants lie outside this region, and therefore could not be replicated. Association with a common variant in LBX1 was present in our study, though its weaker association is presumably due to it being in low linkage equilibrium with the most highly associated SNP near LBX1.

We show the first in vitro evidence for the 391T allele's effect on protein function by overexpression in HEK293 cells. This missense variant is predicted to be deleterious by both Polyphen-2 and SIFT. We observed an effect of the minor allele on manganese transport with little effect on either iron or zinc transport, observations consistent with the variant's prior association with blood manganese levels that we confirmed in our AIS cohort. Moreover, loss of SLC39A8 in mice resulted in decreased manganese concentrations in blood and other tissues, while concentrations of iron and zinc were unaffected[32]. Combined with our genetic data, these results suggest that blood manganese levels could be a contributing factor to scoliosis risk.

Although loss of this gene in mouse is lethal in early embryogenesis, we observed no substantial mortality among either heterozygous or homozygous mutant zebrafish allowing us to assess effects of its loss throughout development and into adulthood. We observed a significantly increased number of spinal abnormalities including thoracic spinal curvature and caudal vertebral fusions in zebrafish with a disruptive mutation within the coding region of the slc39a8 gene including zebrafish produced from heterozygous mothers, suggesting that the phenotype is not maternal zygotic. These abnormalities are present as early as 12 weeks post-fertilization. Unlike col8a1a mutant zebrafish, we observe no notochord defects or issues during spinal

development before 12 weeks, suggesting that rather than an early patterning defect, the spinal fusions and thoracic bunching observed in slc39a8 mutant fish may be due to abnormal growth of bone, abnormal growth of the fish in general or due to locomotor defects during juvenile stages. Although AIS in humans is defined as scoliosis without vertebral fusions, other AIS zebrafish models (col8a1a and kif6) display vertebral fusions due to differences in the spinal development between zebrafish and humans but are considered good models of AIS[20,33]. The structural and functional abnormalities present in slc39a8 mutant fish suggest that the SLC39A8 variant (rs13107325) may contribute to human scoliosis risk through multiple mechanisms, including alterations of spinal growth and/or reduction in motor activity, both of which are consistent with an essential role for manganese as a cofactor for a wide array of enzymes well known to be critical for human health[34]. We observed only mild thoracic curvature in slc39a8 mutant fish.

Our in situ results suggest that slc39a8 may have a role in spinal development directly as it is expressed in the developing zebrafish eye, brain, and spinal cord, but as it is also expressed in endodermal tissue more generally, the effect of gene disruption may be through its expression in the developing liver and its effect on systemic manganese levels.

Given the critical necessity of heavy metal cofactors for a multitude of molecular and cell- biological processes, it is not surprising that the rs13107325 SLC39A8 coding variant is associated with a variety of human traits and disorders, including lipid levels, blood pressure, body-mass index, schizophrenia[15,18,22–25,35], and now as a risk factor underlying severe scoliosis. The combined evidence that the minor allele of this SNP reduces the transport function of this protein and is associated with reduced serum manganese,[15] suggests the possibility of manganese supplementation for the treatment of one or more of these disorders. Importantly, manganese supplementation was recently shown to correct biochemical abnormalities (glycosylation and blood manganese levels) and resulted in tangible clinical improvement in patients with SLC39A8 deficiency with little evidence of toxicity[36]. Future studies are needed to determine whether manganese supplementation may be a therapeutic option to prevent scoliosis progression in at risk individuals.

## Methods

**Subjects, sequencing, and genotyping**. Whole-exome sequence data were generated for 457 AIS European American cases and 987 European American controls for the discovery cohort. For all AIS cases, spinal radiographs were used for diagnosis or Cobb angle was abstracted from clinical records, if unavailable. Exome sequenced cases were included only if Cobb angle > 20. The mean Cobb angle of cases was 52. Controls were not evaluated for scoliosis, and were therefore unselected for presence or absence of scoliosis. Informed consent was obtained for all subjects. IRB approval was obtained for this study by each contributing institution. Exome capture was performed using Agilent All-exome capture kits. Sequencing was performed using paired-end Illumina sequencing to a minimum depth of 30x with >90% of the captured regions covered with at least 8× read depth. Sequencing was performed at the McDonnell Genome Institute at Washington University in St. Louis. Alignment was performed using Novoalign and variant called using SAM-TOOLS. For the expanded control cohort, all non-Finnish European individuals from the Exome Aggregation Consortium (ExAC) were used as control samples ($N = 34,057$). All genotypes, including ExAC genotyping counts, were processed excluding SNPs with call rates < 0.95, minor allele frequency < 0.01 or Hardy–Weinberg equilibrium $P$-values < $10^{-5}$. As a replication cohort, 857 AIS cases and 1095 controls from seven sites (Washington University in St. Louis, Shriners Hospital for Children in St. Louis, University of Iowa, Hospital for Special Surgery, University of Wisconsin, Texas Scottish Rite Hospital and University of Colorado) and were genotyped for rs13107325 using either Affymetrix 6.0 genotyping array or KASPAR genotyping (Supplementary Table 1). A subsample was checked for consistency ($N = 100$) and there was complete concordance between genotyping methods. This was also done for exome sequenced individuals to ensure accuracy and we see 100% concordance in this group as well. (Supplementary Data 1). We performed power calculations using the genetic power

calculator available here: [http://zzz.bwh.harvard.edu/gpc/]. Given our sample sizes, this study had >80% power to detect associations with OR > 1.5 and MAF > 0.07.

**Statistical analysis**. For the discovery exome association analysis, expanded exome analysis, replication analysis and combined analysis, associations between genotype and case/control status were assessed using logistic regression in PLINK[37]. Regional association plots were generated using SNAP [https://www.broadinstitute.org/mpg/snap/]. Association with curvature severity was assessed using linear regression with allele count in R. For associations with height, body-mass index, plasma manganese and plasma iron concentrations, t-tests were performed in R. For height, the difference of measured height from CDC reported averages for age-matched and sex-matched individuals was reported [http://www.cdc.gov/growthcharts/]. Ancestry was verified for all AIS cases and in-house controls using EIGENSTRAT[38] (Supplementary Fig. 6) and individuals were confirmed to be unrelated using IBD estimation in PLINK[37].

**In vitro metal ion uptake assays**. Human *SLC39A8* cDNA was obtained from GE Dharmacon (GenBank® accession number BC012125; Catalog no. MHS6278-202832132) and subcloned into a pCMV-Sport6 vector. To generate the A391T (rs13107325) *SLC39A8*, the QuikChange Lightning site-directed mutagenesis kit (Agilent Technologies) was used to mutate the alanine residue at position 391 to threonine. The mutant construct was subsequently verified by Sanger sequencing. HEK 293T cells (ATCC) were transiently transfected with human *SLC39A8*, the A391T *SLC39A8* variant or empty vector pCMV-Sport6 (Effectene; Qiagen) and incubated for 48 h. Prior to uptake, the cells were washed twice with warm serum-free DMEM medium (for iron uptake) or HBSS medium (for zinc/manganese uptake) and incubated for 1 h in serum-free medium (SFM) containing 2% (w/v) BSA to deplete cells of transferrin and to block the nonspecific binding at 37 °C. For iron uptake, the cells were incubated with 2 μM [$^{59}$Fe]ferric citrate in SFM in the presence of 1 mM L-ascorbate for 2 h at 37 °C followed by three washes of iron chelator solution (1 mM bathophenanthroline sulfonate and 1 mM diethylene-triaminepentaacetic acid) to remove any surface-bound iron. For zinc and manganese uptake, cells were incubated with either 2 μM [$^{65}$Zn]ZnCl$_2$ or 2 μM [$^{54}$Mn] MnCl$_2$ in SFM for 2 h at 37 °C followed by three washes of chelator solution (10 mM ethylenediaminetetraacetic acid). The cells were lysed in buffer containing 0.2 N NaOH and 0.2% (w/v) SDS. Radioactivity (as counts per minute; CPM) was determined by γ-counting, and protein concentration was determined colorimetrically by using the *RC DC* protein assay (Bio-Rad). Transport was normalized to SLC39A8 protein level as measured by western blot. Data are expressed as CPM per mg protein and represent mean ± SE of three independent experiments.

**Western blotting of wild-type and A391T *SLC39A8***. HEK 293T cells were transiently transfected with *SLC39A8* constructs as described above. Cells were washed and lysed in RIPA lysis buffer containing 1× protease inhibitor (Roche). Protein concentration was determined by using the RC DC protein assay (Bio-Rad) prior to western blot analysis. Proteins (15 μg) were mixed with Laemmli buffer and electrophoretically separated on a 7.5% polyacrylamide gel, and transferred to a nitrocellulose membrane. The blots were blocked for 1 h in 5% (w/v) nonfat dry milk in Tris-buffered saline containing 0.1% (v/v) Tween 20 (TBST). The blots were then incubated overnight at 4 °C in blocking buffer containing rabbit anti-hZIP8 (0.05 μg/ml; Sigma; Catalog no. HPA038833) or mouse anti-tubulin (1:10,000; Sigma). The membranes were washed four times for 5 min with TBST and incubated with the appropriate HRP-conjugated secondary antibodies. Immunoreactivity was visualized by using enhanced chemiluminescence (Super-Signal West Pico; Pierce) and the FluorChem E digital darkroom (ProteinSimple).

**CRISPR-mediated *slc39a8* disruption in zebrafish**. Briefly, gRNAs designed to target exon 7 of the *slc39a8* gene were synthesized in vitro and microinjected with codon-optimized Cas9 mRNA into one-cell-stage embryos of the AB* zebrafish strain. F$_0$ founders were identified with Illumina sequencing of F$_1$ embryos. F$_1$ mutants were confirmed by Sanger sequencing. In one line used in this study, a 17-bp insertion and 2-bp deletion was detected on one allele at the *slc39a8* locus. Movement was measured using the Noldus DanioVision and data analyzed using EthoVision XT12 software package. Comparisons of movements were performed using t-tests. Movement was defined as reaching a velocity of 5 mm/s and was measured until fish slowed to 1 mm/s. Approval was obtained for this study by the Washington University institutional review board. For all experiments, both male and female zebrafish were used.

**Alizarin Red staining of zebrafish**. Zebrafish were first fixed in 4% phosphate buffered formalin for 48 h. After rinsing with deionized water, they were dehydrated by in 50% ethanol for 48 h followed by 95% ethanol for 48 h. Fish were then stained using a filtered solution of 20 mg Alcian blue in 30 ml acetic acid and 70 ml 100% ethanol for 24 h. After staining, fish were neutralized in a saturated sodium borate solution for 12 h and bleached in a solution of 15 ml 3% hydrogen peroxide and 85 ml 1% potassium hydroxide for 3 h, cleared in a solution of 35 ml saturated sodium borate and 65 ml deionized water for 7 days, stained for bone using a solution of 1% w/v Alizarin red in 1% potassium hydroxide for 24 h and cleared

again in 3 day intervals using alternating solutions of 35% saturated sodium borate and 1% potassium hydroxide until cleared (>4 weeks). The number of animals stained was determined to ensure adequate power to detect altered rates of skeletal phenotypes in KO vs WT. For each genotype, fish were selected randomly for staining and assessment. Three independent blinded assessors of skeletal phenotypes were used to determine the relationship between genotype and phenotype.

**Whole-mount in situ RNA hybridization (WISH)**. To synthesize RNA probes for *slc39a8* gene, we first amplified 833 bp of *slc39a8* cDNA by RT-PCR using 4 dpf embryos (Supplementary Table 2). The *slc39a8* cDNA fragment was re-amplified by PCR to insert T7 RNA polymerase promoter and subcloned into the pSMART vector (CloneSmart HCKan Blunt Cloning Kit, Lucigen). After sequence confirmation, two plasmids (one for anti-sense and the other for sense *slc39a8* RNA probe) were linearized with *Not*I. From the linearized DNAs, we synthesized both anti-sense and sense RNA probes using a digoxigenin RNA labeling kit (Roche). WISH was performed as described previously[39] with several modifications. We used a higher concentration of Proteinase K (25 μg/ml) and treated embryos for 80 min (from 50 to 80 min PK treatment showed similar results). We used a higher probe concentration (200 μg/ml). Lastly, we added 5% of Dextran sulfate (Sigma-aldrich) to the hybridization mix[40].

## Data availability

AIS case exome sequencing data is available from dbGAP under accession number phs001677.v1.p1. Control exome sequencing data are available from dbGAP under accession numbers phs000101.v4.p1 and phs000572.v7.p4. Affymetrix 6.0 genotyping data are available from dbGAP under accession number phs000658.v1.p1. Additional data used in this study are available from the corresponding authors upon reasonable request.

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

## Acknowledgements

Research reported in this publication was supported by National Institute of Arthritis and Musculoskeletal and Skin Diseases under Award Number R01AR067715, Eunice Kennedy Shriver National Institutes of Child Health and Human Development of the National Institutes of Health under the Award Number P01HD084387, the Marfan Foundation Faculty Grant award #81831, Washington University Institute of Clinical and Translational Sciences grant UL1 TR002345 from the National Center for Advancing Translational Sciences of the National Institutes of Health, Washington University Musculoskeletal Research Center (NIH/NIAMS P30 AR057235), the Eunice Kennedy Shriver National Institute Of Child Health & Human Development of the National Institutes of Health under Award Number U54 HD087011 to the Intellectual and Developmental Disabilities Research Center at Washington University. We thank the Genome Technology Access Center in the Department of Genetics at Washington University School of Medicine for help with genomic analysis. The Center is partially supported by NCI Cancer Center Support Grant #P30 CA91842 to the Siteman Cancer Center and by ICTS/CTSA Grant# UL1RR024992 from the National Center for Research Resources (NCRR), a component of the National Institutes of Health (NIH), and NIH Roadmap for Medical Research. The content is solely the responsibility of the authors and does not necessarily represent the official views of the National Institutes of Health. This study was funded with support from University of Missouri Spinal Cord Injury Research Program, Shriners Hospital for Children, and the Children's Discovery Institute of Washington University and St Louis Children's Hospital, and Hope Center DNA/RNA Purification Core at Washington University School of Medicine. Computations were performed using the facilities of the Washington University Center for High Performance Computing, which were partially funded by NIH grants 1S10RR022984-01A1, and 1S10OD018091-01. We thank patients and their families for their participation, as well as Drs Munish Gupta, Keith Bridwell, Mike Kelly, Scott Luhmann, Brian Kelly, Luke Zebala, Lawrence Lenke and Christi Abeln.

## Author contributions

C.G., M.D., M.K., L.S.-K., R.G., C.W., N.M., P.G., J.M., C.R., M.H., and C.C. supervised the experiments. C.G. and M.D. conceived and designed the study. G.H. and C.G. wrote the manuscript. G.H., K.M., S.J., B.S., L.A., M.N., J.W., Z.U., E.B., and J.S performed the experiments and analyzed the data. All authors discussed the results and reviewed the final manuscript.

## Additional information

**Competing interests:** The authors declare no competing interests.

