## [Peer Review file · Nature Communications]

Reviewers' comments:

Reviewer #1 (Remarks to the Author):

Geller et al. have embarked on whole exome sequencing of a set of 457 severe AIS cases and 987 controls; followed by replication of the top associated SNP (a non-synonymous missense variant rs1310732 with MAF mapping within SLC39A8) in additional 841 cases and 1095 controls, resulting in a combined OR = 1.89; P = 3.64×10^{-25} . In addition, association with several clinical correlates was identified, among which association with lower Mn serum levels in heterozygous carriers of the rare allele, as compared to non-carriers. This is in line with findings of previous GWAS implicating the variant with several conditions, diseases and heavy metal ion levels. The authors go two steps further pursuing functional follow-up in a HEK293 overexpression cell model (to demonstrate an effect at the protein level of Mn transpor) and in a zebrafish model (resulting in a clear vertebral anomalous phenotype analogous to AIS). The findings corroborate a role of SLC39A8 as a membrane transporter responsible for manganese uptake into cells, and implicate a potential role of manganese deficiency in the etiology of AIS.

The paper is well written, and while it can initially be perceived as incremental in relation to previous efforts on AIS genetics, I do think it holds a novel message representing a sound effort to take a genetic discovery into a clinical context by means of functional follow-up.

Below several comments about the paper:

1. The inclusion of the ex-AC extended set of controls is confusing. Of course the P-value becomes stronger by including a larger set of controls from a population with a bit lower MAF. Yet, this opens many more questions about how adequate are bringing controls from a different population-base. I suggest just describing the MAF in that independent setting (ex-AC) to support the robustness of the MAF in controls from the study setting. Otherwise, I would like to see genomic inflation factors and a graphic illustration of genomic principal components to see that cases and controls have a decent overlap in those dimensions (and not a result of stratification).
2. Only 50K markers were analyzed in a massive effort. Why did the authors filter out variants with MAF < 0.01? The authors are missing an enormous amount of variants of considerable deleterious potential which is a pity considering the scale of this sequencing effort. Will that scrutiny be part of another paper? Also losing 15K markers to include the ex-AC just to pimp-up the significance is silly.
3. Regarding the MAF please describe it early on in the manuscript (abstract) and a MAF of 0.07-0.14 is far from rare and even not less frequent., so please refrain from using this along the manuscript (including title) and emphasize the "coding" properties of the variant(s).
4. Discuss why other efforts in Europeans did not find this variant in the context of power, imputation difficulty or presence in genotyping arrays. If results of previous efforts in Europeans are available please look-up and report the P-value.
5. The authors do not describe how population stratification was controlled for in the two settings (discovery and replication).
6. The authors occasionally employ genetic terminology in an inaccurate way that can be confusing to the reader:
 - a. This study is not a GWAS, it is a whole-exome sequencing effort followed by replication genotyping so please amend Table 1.

- b. When referring to a SNP not being present in a population, rather refer to the SNP as being “monomorphic” in a given population.
- c. A SNP (rs1310732) is not associated per se with lower/higher levels of a trait, when direction is described please include the specific allele for which the change in the trait is presented.
7. In the methods section “statistical analysis” the analysis in PLINK for the case/control association testing should be “logistic” regression instead of “linear” regression.
8. Please specify if the OR is per allele or per genotype, as an OR of 2x increased risk per allele seems a very large (unreal) effect. Please discuss a potential “winners curse” bias of the effect estimate and report the de-novo genotyping results in this context (see comment below).
9. The authors refer to regional association plots created with SNAP, but I do not find them or any reference to them along the manuscript.
10. A figure illustrating the position of the variant within the gene (exons) will be informative. Regarding the identified variant please make a thorough analysis and description of its annotation and potential deleterious properties (or of variants in the vicinity) using tools like PolyPhen-2, SIFT, etc.
11. There is no description of the methods use to diagnose scoliosis. Were radiographs and measurement of Cobb angles performed in both cases and controls of the discovery and replication settings? Was it performed by radiologists in all cases or retrieved from patient records? Regarding this, Figure 2A should be done on “cases only” to examine if the genotypes are associated with severity. Including the controls is just a proxy of the case control status definition driving the discovered association. The analysis of height and BMI (Figures 2b and 2c) should be presented for all three genotype groups and report a P for trend (instead of regrouping/changing to a dominant genetic model). Otherwise, correct for multiple model testing. Related to the latter, it is not clear why the analysis of Mn and Fe levels is done on the heterozygotes and non-carriers. Please explain the design resulting into this. Why not measure this in the homozygous?
12. The genotyping of the replication cohort is very ill defined. The authors state cases and controls come from 7 different centers and use either an in-silico (GWAS-look-up in imputed data) or de-novo genotyping (KASPAR). Please indicate how many/which centers used each genotyping technique and describe the number of cases and controls per center.

Reviewer #2 (Remarks to the Author):

In the manuscript titled "A low-frequency missense variant in SLC39A8 is associated with severe idiopathic scoliosis" by Haller et al. the authors report the identification of a low frequency missense mutation in SLC39A8 associated with severe AIS, low Mn²⁺ blood levels and short stature. They then went ahead and assayed the mutation in vitro in HEK293 cells, showing a small but significant defect in Mn²⁺ uptake. To validate the target in vivo they generated a zebrafish mutant and observed locomotion deficits in 4 pdf larvae, which were ameliorated by Mn²⁺ supplementation, and spine defects in adult animals.

Altogether, this manuscript reports interesting findings on the function of poorly known transporter in a common spine defect (AIS) of complex etiology whose heritability is largely undefined at this point. Therefore, these findings are significant for the field of spine biology and human genetics. It is important to note that the identified genetic variants originate from a whole exome sequencing

approach that was validated in vivo, which provides good support for the conclusions. There is however a need for a more in-depth characterization of the slc38a9 in vivo model as several key data are missing and other points need further clarification.

Major points:

1-The authors report motion defects in 4 dpf larvae and spine defects in 9 mpf fish. 1A-What is the earliest spine defects are detected? At nine months zebrafish are not old, but are well beyond what can be considered as a late onset adult phenotype (about 3 mpf), raising some concern that the observed phenotype might be related to premature aging. 1B-Are the fish zygotic or maternal-zygotic (mz) mutants? 1C-If the latter, is the mz phenotype stronger?

2-The skeletal preps in Figure 4 are of good quality, but it is hard to tell whether there is lateral curvature. It would be best if the authors could: 2A: show dorsal views; 2B-provide a supplemental figure with additional examples.

3-The authors report motion defects in 4 dpf larvae and shortened body length at 9 mpf. 3A-Is axis elongation impaired? (i.e. body length at 24 hpf). 3B-Do 4 or 5 dof larvae show body length defects?

4A-Where is slc38a9 expressed at 4dpf? (i.e. in situ hybridization) 4B-Is there any expression data from mouse or humans that might be cited?

5-If spine defects can be detected relatively early, It would be very interesting to know if the phenotype can be rescued by Mn²⁺ supplementation.

Minor points:

6-In the published reports of human mutations in SLC38A9, did all the affected individuals present scoliosis? Perhaps a more detailed discussion is warranted.

7-Past the 3 dpf time point zebrafish should be referred to as larvae and not embryos.

Reviewer #3 (Remarks to the Author):

The study identified a highly significant association between SNP rs13107235 and severe adolescent idiopathic scoliosis. The study was carefully conducted and included detailed functional study.

I recommend its publication without any major changes.

Reviewer #4 (Remarks to the Author):

Haller et al conducted an exome-wide association study for adolescent idiopathic scoliosis (AIS), and found that a missense SNP rs13107325 was associated with severe AIS at a combined p value of 3.64×10^{-25} (OR=1.89). Functional studies demonstrated that cells transfected with SLC39A8 increased influx of cations. The transporter had the greatest effect on Mn²⁺, with overexpression increasing Mn²⁺ uptake by more than 10-fold. Cells expressing the SLC39A8 p.A391T allele transported less Mn²⁺ than wild-type. Furthermore, slc39a8 mutant zebrafish had vertebral

abnormalities, impaired growth, and decreased motor activity compared to wild-type zebrafish.

Major concerns:

1, The statistical methods are not clear to me, I guess you applied Logistic regression for the case-control study, right? Did you mention this in your methods? What co-variables were included in your Logistic model? Are the confounders such as height, body mass index, plasma manganese and plasma iron concentrations considered?

Especially, SNP rs13107325 is associated with height, body mass index, plasma manganese and plasma iron concentrations, could you clarify the effect on AIS is because of SNP rs13107325 or any other factors? I would suggest the authors perform a Mendelian Randomization analysis, because SNP rs13107325 is confirmed to associated with BMI in many other studies.

Therefore, the association results presented here should be interpreted carefully.

In your methods, you said "Association with curvature severity was assessed using linear regression with allele count in R". What was the dependent variable and what were the co-variables? Where did you present these results?

You also said "For associations with height, body mass index, plasma manganese and plasma iron concentrations, t-tests were performed in R". Where were the association results for SNP rs13107325 with height, body mass index, plasma manganese and plasma iron concentrations in your cohort? Figure 2 is the case only analysis.

2, How you define SEVERE AIS? How the 457 AIS European American cases and 987 European American controls were recruited, what were the criteria for cases and controls? Similarly, what were the criteria for the replication samples? Did you test the population structure of the samples? Is there any inflation in you study samples?

What is the power for this exome wide association study? Is it enough to discover a low frequency SNP in ~450 samples?

3, Could you please make a table to present the association results of the previous reported loci in your cohort? Did you do any conditional analysis?

4, You mentioned Imputation in your Methods, did you use any imputation data in your association analysis? What is the reference panel you applied for your imputation?

What was the allele frequency distribution of the 52,480 SNPs? Did you compare them with 1000 Genomes and ExAC?

5, Are you sure SNP rs13107325 is not presented in the ref6,7,12,33, as you discussed that "Interestingly, the two AIS associated SNPs found in this study are not present among Asian or African populations, explaining the lack of association in prior AIS genome-wide association studies6,7,12,33."

Minor ones:

Usually we defined low-frequency allele when the minor allele frequency is great than 1% and less than 5%, by this definition, the minor allele frequency of SNP rs13107325 is ~6%, it is common. I would suggest you change the title of the manuscript.

Where the sequencing was done? What is the quality control of the sequencing?

For the combined results, you combined the AIS GWAS, AIS GWAS Expanded and AIS replication, right? How you did the combination?

In the Legend of Figure 2, "Clinical characteristics of rs13107325 carriers", it is better saying "Clinical characteristics of AIS cases with different rs13107325 genotypes".

Ref8 and ref11 were duplicated.

Could you please add the 95%CI in the Abstract for the combined effect OR 1.89.

Delete ")" at the end of line 109.

Responses to Reviewers' comments:

Reviewer #1:

1. The inclusion of the ex-AC extended set of controls is confusing. Of course the P-value becomes stronger by including a larger set of controls from a population with a bit lower MAF. Yet, this opens many more questions about how adequate are bringing controls from a different population-base. I suggest just describing the MAF in that independent setting (ex-AC) to support the robustness of the MAF in controls from the study setting. Otherwise, I would like to see genomic inflation factors and a graphic illustration of genomic principal components to see that cases and controls have a decent overlap in those dimensions (and not a result of stratification).

Response: We agree with the reviewer and have now included a qq-plot and genomic inflation factor ($\lambda=1.09$) for the exome-wide analysis of our AIS cases vs ExAC as controls (Supplementary Figure 1) In this analysis, the only genome-wide significant SNP is rs13107325. Because individual genotypes are not publicly available for ExAC, we used genotype counts for this dataset which are available for all coding variants.

2. Only 50K markers were analyzed in a massive effort. Why did the authors filter out variants with $MAF < 0.01$? The authors are missing an enormous amount of variants of considerable deleterious potential which is a pity considering the scale of this sequencing effort. Will that scrutiny be part of another paper? Also losing 15K markers to include the ex-AC just to pimp-up the significance is silly.

Response: We appreciate the reviewers acknowledgement of this effort, and note that variants were included in previous rare variant burden studies from our laboratory but because variants with MAF less than 0.01 have nearly no power for single SNP analyses with our sample size they were excluded from the current association study of single variants. All common variants sequenced in the initial exome sequencing were included in the primary analysis. Because ExAC contains only variants within the exome, we had to use only the intersection of variants present in both ExAC and AIS cases in the "extended" association analysis of AIS cases vs ExAC. The results are very similar if we take all genotyped SNPs.

3. Regarding the MAF please describe it early on in the manuscript (abstract) and a MAF of 0.07-0.14 is far from rare and even not less frequent., so please refrain from using this along the manuscript (including title) and emphasize the "coding" properties of the variant(s).

We agree with the reviewer, and added a sentence describing the frequencies in various populations to the abstract and removed the word "low-frequency" from the title. We added the word "functional", however, as our in vitro data demonstrates an effect of the variant on protein function.

4. Discuss why other efforts in Europeans did not find this variant in the context of power, imputation difficulty or presence in genotyping arrays. If results of previous efforts in Europeans are available please look-up and report the P-value.

We believe the largest factor contributing to this association not being found previously is the severity of the scoliosis in our cohort. This is described in the discussion: "In addition, our cohort is composed mostly of AIS cases with severe scoliosis, possibly making the genetic load in this cohort higher than cohorts used in previous studies." We do not believe the p-values for the associated SNP (or SNPs in high LD with it) are publicly available.

5. The authors do not describe how population stratification was controlled for in the two settings (discovery and replication).

We added the following sentence to the statistical analysis section of the Methods: “Ancestry was verified for all AIS cases and in-house controls using EIGENSTRAT³⁸.”

6. The authors occasionally employ genetic terminology in an inaccurate way that can be confusing to the reader:

a. This study is not a GWAS, it is a whole-exome sequencing effort followed by replication genotyping so please amend Table 1.

We apologize for the confusing use of terminology, and have clarified by changed the terminology as suggested to whole-exome association study where GWAS appeared previously.

b. When referring to a SNP not being present in a population, rather refer to the SNP as being “monomorphic” in a given population.

We agree and changed the wording in the abstract and the discussion (line 187): “Interestingly, the two AIS associated SNPs are monomorphic in Asian or African populations, explaining the lack of association in prior AIS genome-wide association studies, largely in Asian populations”.

c. A SNP (rs13107325) is not associated per se with lower/higher levels of a trait, when direction is described please include the specific allele for which the change in the trait is presented.

We agree with the reviewer, and modified the abstract to read: “Clinically, the minor allele of rs13107325 was associated with greater spinal curvature, decreased height, increased BMI and lower plasma manganese concentration in our AIS cohort” and a sentence in the results (line 123): “We observed a dose-dependent correlation between SLC39A8 391T allele count and scoliosis curve severity as measured by the Cobb angle (Figure 2A)”.

7. In the methods section “statistical analysis” the analysis in PLINK for the case/control association testing should be “logistic” regression instead of “linear” regression.

We agree and have changed the sentence.

8. Please specify if the OR is per allele or per genotype, as an OR of 2x increased risk per allele seems a very large (unreal) effect. Please discuss a potential “winners curse” bias of the effect estimate and report the de-novo genotyping results in this context (see comment below).

For SNPs that are less frequent than most reported in GWAS generally, as this one is, an OR per allele of 2 is not that unusual. The SNP is twice as frequent in cases than controls. This would be nearly impossible for a SNP whose allele frequency was near 50%, but 7% vs 14%, although only a 7% difference, is 2x as frequent.

9. The authors refer to regional association plots created with SNAP, but I do not find them or any reference to them along the manuscript.

We reviewer is correct and this is not shown, and we have now removed the reference.

10. A figure illustrating the position of the variant within the gene (exons) will be informative. Regarding the identified variant please make a thorough analysis and description of its annotation and potential deleterious properties (or of variants in the vicinity) using tools like PolyPhen-2, SIFT, etc.

We added figure showing the location of the SNP within the gene (Supplementary Figure 2). We also added a sentence describing the annotations by PolyPhen-2 and SIFT (both predict it to be deleterious): “This missense variant is predicted to be deleterious by both Polyphen-2 and SIFT.”

11. There is no description of the methods use to diagnose scoliosis. Were radiographs and measurement of Cobb angles performed in both cases and controls of the discovery and replication settings? Was it performed by radiologists in all cases or retrieved from patient records? Regarding this, Figure 2A should be done on “cases only” to examine if the genotypes are associated with severity. Including the controls is just a proxy of the case control status definition driving the discovered association. The analysis of height and BMI (Figures 2b and 2c) should be presented for all three genotype groups and report a P for trend (instead of regrouping/changing to a dominant genetic model). Otherwise, correct for multiple model testing. Related to the latter, it is not clear why the analysis of Mn and Fe levels is done on the heterozygotes and non-carriers. Please explain the design resulting into this. Why not measure this in the homozygous?

We have now added a section to the methods describing the inclusion criteria for AIS cases used in this study. Spinal radiographs were used to diagnose cases and abstracted from clinical records as the radiologists at our center do not typically measure Cobb angles. Cases were included only if Cobb angles >20. The mean Cobb angle of cases was 52. Controls were not evaluated for scoliosis, and were unselected for presence or absence of scoliosis.

We apologize that Figure 2 was unclear, as only cases were used for all sections of this figure. We changed the analysis of Cobb angle to be consistent with the other analyses and performed a dominant test due to the small number of cases with Thr/Thr genotype. We now report only data from heterozygotes and non-carriers for all clinical variables because we have too few individuals who are homozygous for the rare allele.

12. The genotyping of the replication cohort is very ill defined. The authors state cases and controls come from 7 different centers and use either an in-silico (GWAS-look-up in imputed data) or de-novo genotyping (KASPAR). Please indicate how many/which centers used each genotyping technique and describe the number of cases and controls per center.

We have included a supplementary table (Supplementary Table 1) describing the samples and genotyping used. No imputed data was used to determine the genotype of the lead SNP in any cohort.

Reviewer #2 (Remarks to the Author):

Major points:

1-The authors report motion defects in 4 dpf larvae and spine defects in 9 mpf fish. 1A-What is the earliest spine defects are detected? At nine months zebrafish are not old, but are well beyond what can be considered as a late onset adult phenotype (about 3 mpf), raising some concern that the observed phenotype might be related to premature aging. 1B-Are the fish zygotic or maternal-zygotic (mz) mutants? 1C-If the latter, is the mz phenotype stronger?

We can detect the spine defects as early as 12 weeks. We have now stained 12 week old fish with alizarin red (1/4 of heterozygote x heterozygote offspring) and see the same relative frequency of spinal anomalies as in the 9 month old fish previously (Supplementary Figure 4). The fish in Figure 1C are maternal-zygotic mutant as they were the product of a homozygote by homozygote cross. There does not appear to be a stronger phenotype in maternal-zygotic mutants.

2-The skeletal preps in Figure 4 are of good quality, but it is hard to tell whether there is lateral curvature. It

would be best if the authors could: 2A: show dorsal views; 2B-provide a supplemental figure with additional examples.

We have not observed any lateral curvature, therefore no photos were shown. We have now added a supplemental figure with more examples of skeletal abnormalities in the *slc39a8* ^{-/-} fish (Supplementary Figure 3 (for 9 month old zebrafish) and Supplementary Figure 4 (for 12 week old zebrafish)).

3-The authors report motion defects in 4 dpf larvae and shortened body length at 9 mpf. 3A-Is axis elongation impaired? (i.e. body length at 24 hpf). 3B-Do 4 or 5 dof larvae show body length defects?

We see no defects in length at 24 hpf or at 5dpf. The growth difference only become apparent at later stages of development.

4A-Where is *slc38a9* expressed at 4dpf? (i.e. in situ hybridization) 4B-Is there any expression data from mouse or humans that might be cited?

We performed whole-mount in situ hybridizations in 4 day post-fertilization zebrafish embryos and observed mRNA expression of *slc39a8* in the eye, spinal cord and in endodermal tissues, including the liver.

5-If spine defects can be detected relatively early, It would be very interesting to know if the phenotype can be rescued by Mn²⁺ supplementation.

We agree that these experiments will be very exciting, however, at the current time, we cannot perform the long-term treatment that this would entail within our zebrafish facility.

Minor points:

6-In the published reports of human mutations in SLC38A9, did all the affected individuals present scoliosis? Perhaps a more detailed discussion is warranted.

Unfortunately, scoliosis is a phenotype that is not always reported or investigated in children with developmental disabilities, so while some of the human patients with *SLC39A8* homozygous mutations were noted to have scoliosis, it is was unlikely to have been investigated in all children particularly those who are young.

7-Past the 3 dpf time point zebrafish should be referred to as larvae and not embryos.

Thank you for the correction. We have altered the text where appropriate.

Reviewer #3 (Remarks to the Author):

The study identified a highly significant association between SNP rs13107235 and severe adolescent idiopathic scoliosis. The study was carefully conducted and included detailed functional study.

I recommend its publication without any major changes.

Reviewer #4 (Remarks to the Author):

Major concerns:

1, The statistical methods are not clear to me, I guess you applied Logistic regression for the case-control

study, right? Did you mention this in your methods? What co-variables were included in your Logistic model? Are the confounders such as height, body mass index, plasma manganese and plasma iron concentrations considered?

Especially, SNP rs13107325 is associated with height, body mass index, plasma manganese and plasma iron concentrations, could you clarify the effect on AIS is because of SNP rs13107325 or any other factors? I would suggest the authors perform a Mendelian Randomization analysis, because SNP rs13107325 is confirmed to associated with BMI in many other studies.

Therefore, the association results presented here should be interpreted carefully.

In your methods, you said "Association with curvature severity was assessed using linear regression with allele count in R". What was the dependent variable and what were the co-variables? Where did you present these results?

You also said "For associations with height, body mass index, plasma manganese and plasma iron concentrations, t-tests were performed in R". Where were the association results for SNP rs13107325 with height, body mass index, plasma manganese and plasma iron concentrations in your cohort? Figure 2 is the case only analysis.

We performed logistic regression for the primary association analysis, as described in the methods section. We did not include any covariates in the initial analysis, because we only measured plasma manganese and iron levels in a limited number of patients to corroborate the previous association of this variant with these traits in our cohort. In addition, we do not have height or BMI measured for our control cohorts. The association analysis for BMI, height, curve, blood Mn and Fe was done with cases only. The results from this are presented in Figure 2 and reported in the Results section labeled "Effect of SLC39A8 A391T on Clinical Characteristics".

2, How you define SEVERE AIS? How the 457 AIS European American cases and 987 European American controls were recruited, what were the criteria for cases and controls? Similarly, what were the criteria for the replication samples? Did you test the population structure of the samples? Is there any inflation in you study samples?

We define severe AIS as curves >40 degrees which is the degree of curvature which is typically treated surgically. We have now added a section to the methods describing the inclusion criteria for AIS cases. Spinal radiographs were used to diagnose cases and abstracted from clinical records as the radiologists at our center do not typically measure Cobb angles. Cases were included only if Cobb angles >20. The mean Cobb angle of cases was 52. Controls were not evaluated for scoliosis, and were unselected for presence or absence of scoliosis.

We added the following sentence to the statistical analysis section of the Methods: "Ancestry was verified for all AIS cases and in-house controls using EIGENSTRAT³⁸."

What is the power for this exome wide association study? Is it enough to discover a low frequency SNP in ~450 samples?

With the extended controls from ExAC, we have 87% power to detect an association ($\alpha=5 \times 10^{-8}$) with $OR=2$ and a frequency of 7%. Given our sample size of AIS cases and in-house controls only, we have 40% power to identify a SNP with an odds ratio of 2 down to a frequency of of 7% (as is rs13107325 in controls).

3, Could you please make a table to present the association results of the previous reported loci in your cohort? Did you do any conditional analysis?

As we used exome sequence data in this project, we do not have genotypes for the majority of previously reported SNPs as most are intronic or intergenic. We do see an association between a SNP near GPR126, a previously association locus with AIS, but the association is not very significant (0.001 in our analysis), probably due to the low LD ($r^2=0.62$ in Europeans) between the sequenced variant (rs11155242) and the previously reported SNP (rs6570507). No SNPs in LD ($R^2>0.5$) with the top SNP from L BX1 or near BNC2 were present in our association analysis due to none of them being coding SNPs.

4, You mentioned Imputation in your Methods, did you use any imputation data in your association analysis? What is the reference panel you applied for your imputation?

We do not report any imputation results so we have removed that sentence from the methods section. All association results use genotyped (not imputed) SNPs.

What was the allele frequency distribution of the 52,480 SNPs? Did you compare them with 1000 Genomes and ExAC?

Part of our analysis was performing an “extended” association with all SNPs called from exome sequencing in our cohort using ExAC as the control cohort. The only SNP to remain significant in this analysis was rs13107325. Therefore, the allele frequency distribution in our sample is largely similar to that of ExAC as we would see massive p-value inflation if that were not the case. We have now included a Q-Q plot for the comparison of our AIS cases and ExAC controls (Supplementary Figure 1).

5, Are you sure SNP rs13107325 is not presented in the ref 6,7,12,33, as you discussed that “Interestingly, the two AIS associated SNPs found in this study are not present among Asian or African populations, explaining the lack of association in prior AIS genome-wide association studies 6,7,12,33.”

The cited studies used cohorts of Asian descent in their primary analyses. The SNP rs13107325 is monomorphic in Asian populations. The SNP was likely genotyped in these studies but would not have been associated due to frequency in the primary populations used. The meta-analysis analyzed L BX1 variants only, although it included European cohorts as well.

Minor ones:

Usually we defined low-frequency allele when the minor allele frequency is great than 1% and less than 5%, by this definition, the minor allele frequency of SNP rs13107325 is ~6%, it is common. I would suggest you change the title of the manuscript.

We agree with the reviewer and have changed the manuscript to no longer refer to the associated SNP as “low frequency”.

Where the sequencing was done? What is the quality control of the sequencing?

Sequencing was performed at the McDonnell Genome Institute at Washington University in St. Louis. We have added the following to the methods section of the manuscript: “Sequencing was performed using paired-end Illumina sequencing to a minimum depth of 30x with >90% of the captured regions covered with at least 8x read depth.”

For the combined results, you combined the AIS GWAS, AIS GWAS Expanded and AIS replication, right? How you did the combination?

We performed an analysis simply combining the samples and performing one test of association. The AIS GWAS and AIS GWAS Expanded use different control cohorts, but the same AIS cases. We did not perform a meta-analysis.

In the Legend of Figure 2, "Clinical characteristics of rs13107325 carriers", it is better saying "Clinical characteristics of AIS cases with different rs13107325 genotypes".

We agree with the reviewers and have changed the title of the figure.

Ref8 and ref11 were duplicated.

Thank you for catching this error. We removed the duplication.

Could you please add the 95%CI in the Abstract for the combined effect OR 1.89. Delete ")" at the end of line 09.

We have added the 95% CI.

Reviewers' comments:

Reviewer #1 (Remarks to the Author):

I thank the authors for their efforts to revise the manuscript, which is substantially improved after revision. Nevertheless, I still have a few points of concern:

1. Title: Rather than "functional" I recommend using the term "deleterious"
2. The inclusion of the ExAC control set remains extremely confusing and unnecessary. The abstract reads very well with a clear description of the discovery and replication set, but then reports the analysis of the ExAC in the combined analysis ($P = 3.64 \times 10^{-25}$; $OR = 1.89$ ($CI=1.67-2.14$)). Other than window dressing significance there is no reason why this strategy adds to the manuscript. The authors failed to provide plots of the principal components to show that cases and controls are adequately matched and balanced. Please remove the ExAC analysis from the association analysis and see my subsequent point. Was there any level of relatedness among (sets of) cases?
3. Typical exome-sequencing efforts use large reference sets (i.e. ExAC) to prune out variants that are unlikely to explain the phenotype under the principle that causative variants will not be seen in the reference set. This type of analysis is relevant to rare variants (giving rise to my previous question about scrutinising variants with a MAF below 0.01) and helps to prioritise those variants worthy of functional follow-up. Please examine a few exome sequencing papers which have been successful findings variants using this approach and chose to either run such analysis in your data and/or discuss the difference with the plain association approach employed.
4. There is a typo on line 99 of page 3 with the name of the SNP: rs13107235 should be rs13107325. Please clarify in the Supp Figure 1 that the repeated annotation of rs13107325 from mGWAS is in relation to other traits (i.e., BMI, BP and psychiatric traits, etc) .
5. Please discuss about the effect size of the association. Typical complex traits have much smaller effect sizes as demonstrated by large well-powered GWAS. I still believe "winners course" is still an important source of the large effect size. Please do so in the discussion section providing arguments against or in favour of that contingency.

Reviewer #2 (Remarks to the Author):

The authors have addressed several important issues and have made significant improvements to their manuscript which is of significant interest.

There are however there are two issues that need to be clarified. These may be addressed through editorial changes.

1-The spine defects associated with the SLC39a8 missense variant is severe AIS. Whereas, in the loss of function zebrafish model vertebral defects but no kinking of the axis (e.g. such as in col8a1a mutants) are observed. Therefore, these disparate results should be discussed. For example, the authors may mention that based on the animal model results it is possible that early vertebral defects may lead to kinking of the spine axis during adolescence. It is also possible that locomotion defects also contribute to abnormal vertebral growth.

2-The issue of timing is still problematic and should be addressed more precisely and linked to the

point made above.

The phenotype observed in *slc39a8* mutant fish is basically vertebral fusions. These are detected at 3 months but are likely present from the time of vertebral patterning, which in zebrafish occurs in a long window between 5 and 21 dpf (see PMID29466731, 29650589, 29624170). As shown in those recent papers, defects in notochord segmentation can cause vertebral fusions similar to those seen in the *slc39a8* fish.

On the other hand, while the *col8a1a* mutation shows a very interesting notochord and spine phenotype that includes vertebral defects, it is associated with kinks in the notochord and spine axis that are present from the embryonic stage and are therefore not of late-onset. Late onset (after 30 dpf) vertebral over growth can cause apparent fusions (see PMID 18927155), it does not seem to be the case in *slc39a8* fish. Therefore, a relatively early vertebral patterning defect is likely the primary cause of the phenotype present in *slc39a8* mutant fish.

Additional minor corrections:

mpf and dpf should be spelled out the first time they are mentioned

Line 168: ...wild-type embryos... should be larvae

Line 169: ...in wild-type 4 dpf embryos... Should say 4 dpf larvae

Figure 5: the legend also refers to 4 dpf embryos and should be larvae.

Reviewer #4 (Remarks to the Author):

The authors have responded to part of the reviewers' concerns, however, more efforts should be made to improve the manuscript.

As the authors described, SNP rs13107325 was associated with height, body mass index, plasma manganese and plasma iron concentrations, I am wondering if the association (between rs13107325 and scoliosis) was a mediation effect by the association between rs13107325 and manganese? or a mediation effect by other associations? Mendelian Randomization analysis might explain casual association.

Because you didn't include any co-variables in your Logistic model (not all of them have these data), the association should be interpreted carefully, and the limitation should be presented.

I am wondering which dataset you were using to generate Supplementary Figure 2, there was a significant SNP at p value of $\sim 10^{-9}$, however, this SNP was presented neither in table 1 nor in figure 1. What did the red line mean? if you combined some datasets, how about the second significant SNP in the figure 1?

Not all of the cases were severe idiopathic scoliosis (Cobb angle $>40^\circ$), in your exome-wide study, the range was 10-180, in your replication study, the range was 20-104.

In the Methods, "Cases were included only if Cobb angle >20 ", but in the Results, the range was 10-180.

"Ancestry was verified for all AIS cases and in-house controls using EIGENSTRAT", please show the

results in supplementary figures.

"A subsample was checked for consistency (N=100) and there was complete concordance between genotyping methods", please show the results in supplementary tables.

In discussion, this SNP is not a rare variant, it is common. "We identified a highly significant association between a moderately rare coding variant..."

Please refer figure 3 in the paragraph "In Vitro Functional Testing".

In line 126, it should be figure 2, right?

Reviewers' comments:

Reviewer #1 (Remarks to the Author):

I thank the authors for their efforts to revise the manuscript, which is substantially improved after revision. Nevertheless, I still have a few points of concern:

1. Title: Rather than "functional" I recommend using the term "deleterious"

Response: Deleterious implies a negative effect. Because this SNP affects so many traits, saying the SNP itself is deleterious would be inaccurate. Therefore, to negate this possibility, we changed the title to describe the SNP as "missense," which is both more neutral and accurate.

2. The inclusion of the ExAC control set remains extremely confusing and unnecessary. The abstract reads very well with a clear description of the discovery and replication set, but then reports the analysis of the ExAC in the combined analysis ($P = 3.64 \times 10^{-25}$; $OR = 1.89$ ($CI=1.67-2.14$)). Other than window dressing significance there is no reason why this strategy adds to the manuscript. The authors failed to provide plots of the principal components to show that cases and controls are adequately matched and balanced. Please remove the ExAC analysis from the association analysis and see my subsequent point. Was there any level of relatedness among (sets of) cases?

Response: We agree with the reviewer, and have removed the ExAC frequencies from the association analysis including and have instead simply stated that the frequency in ExAC and now GNOMAD are consistent with those seen in the controls: "We additionally compared the allele frequency in AIS cases to a large control cohort, consisting of individuals of European descent (Non-Finnish) from the Exome Aggregation Consortium (ExAC) (N=34,057) and observed an allele frequency at rs13107325 consistent with that of controls (MAF= 0.067)."

We have also added a sentence to the methods describing the fact that we ran IBD estimates for our cohorts of AIS cases and controls and do not see any substantial relatedness: "Ancestry was verified for all AIS cases and in-house controls using EIGENSTRAT³⁷ (Supplementary Fig. 6) and individuals were confirmed to be unrelated using IBD estimation in PLINK."

3. Typical exome-sequencing efforts use large reference sets (i.e. ExAC) to prune out variants that are unlikely to explain the phenotype under the principle that causative variants will not be seen in the reference set. This type of analysis is relevant to rare variants (giving rise to my previous question about scrutinizing variants with a MAF below 0.01) and helps to prioritize those variants worthy of functional follow-up. Please examine a few exome sequencing papers which have been successful findings variants using this approach and chose to either run such analysis in your data and/or discuss the difference with the plain association approach employed.

Response: We have already performed rare variant association studies on this data, which were previously published and referenced (Haller et al., and Buchan et al.) and are therefore beyond the scope of the current common variant association analysis.

4. There is a typo on line 99 of page 3 with the name of the SNP: rs13107235 should be rs13107325. Please clarify in the Supp Figure 1 that the repeated annotation of rs13107325 from mGWAS is in relation to other traits (i.e., BMI, BP and psychiatric traits, etc).

Response: We have fixed the typo and have added a sentence to the figure legend: "SNPs are listed once for each independent genome-wide association publication in which they are listed."

5. Please discuss about the effect size of the association. Typical complex traits have much smaller effect sizes as demonstrated by large well-power GWAS. I still believe "winners course" is still an important source of the large effect size. Please do so in the discussion section providing arguments against or in favour of that contingency.

Response: We added the following sentence to the discussion: “This association was replicated in a second independent cohort of AIS patients and controls. While we observed an odds ratio near 2 in both the discovery cohort and replication cohort, this is uncharacteristically large for genome-wide association studies. The effect size of the SNP may be overestimated, however, justifying further study in additional cohorts of AIS patients.”

Reviewer #2 (Remarks to the Author):

The authors have addressed several important issues and have made significant improvements to their manuscript which is of significant interest.

There are however there are two issues that need to be clarified. These may be addressed through editorial changes.

1-The spine defects associated with the SLC39a8 missense variant is severe AIS. Whereas, in the loss of function zebrafish model vertebral defects but no kinking of the axis (e.g. such as in col8a1a mutants) are observed. Therefore, these disparate results should be discussed. For example, the authors may mention that based on the animal model results it is possible that early vertebral defects may lead to kinking of the spine axis during adolescence. It is also possible that locomotion defects also contribute to abnormal vertebral growth.

2-The issue of timing is still problematic and should be addressed more precisely and linked to the point made above.

The phenotype observed in slc39a8 mutants fish is basically vertebral fusions. These are detected at 3 months but are likely present from the time of vertebral patterning, which in zebrafish occurs in a long window between 5 and 21dpf (see PMID29466731, 29650589, 29624170). As shown in those recent papers, defects in notochord segmentation can cause vertebral fusions similar to those seen in the slc39a8 fish.

On the other hand, while the col8a1a mutation shows a very interesting notochord and spine phenotype that includes vertebral defects, it is associated with kinks in the notochord and spine axis that are present from the embryonic stage and are therefore not of late-onset. Late onset (after 30 dpf) vertebral over growth can cause apparent fusions (see PIMD 18927155), it does not seem to be the case in slc39a8 fish. Therefore, a relatively early vertebral patterning defect is likely the primary cause of the phenotype present in slc39a8 mutant fish.

Response: We have address this, we added the following to the results: “Often the thoracic malformations involved vertebral bunching leading to mild spinal curvature.” The discussion was modified to read: “These abnormalities are present as early as 12 weeks post-fertilization. Unlike col8a1a mutant zebrafish, we observe no notochord defects or issues during spinal development before 12 weeks, suggesting that rather than an early patterning defect, the spinal fusions and thoracic bunching observed in *slc39a8* mutant fish may be due to abnormal growth of bone, abnormal overall growth of the fish, or due to locomotor defects during adolescence. “ We have also added a supplementary figure (Supplementary Fig. 5) showing calcein staining of zebrafish at 13 days post-fertilization showing no apparent spinal abnormalities at that age for *slc39a8* homozygous mutant zebrafish.

Additional minor corrections:

mpf and dpf should be spelled out the first time they are mentioned

Line 168: ...wild-type embryos... should be larvae

Line 169: ...in wild-type 4dpf embryos... Should say 4dpf larvae

Figure 5: the legend also refers to 4 dpf embryos and should be larvae.

Response: We have corrected these errors.

Reviewer #4 (Remarks to the Author):

The authors have responded to part of the reviewers' concerns, however, more efforts should be made to improve the manuscript.

As the authors described, SNP rs13107325 was associated with height, body mass index, plasma manganese and plasma iron concentrations, I am wondering if the association (between rs13107325 and scoliosis) was a mediation effect by the association between rs13107325 and manganese? or a mediation effect by other associations? Mendelian Randomization analysis might explain casual association.

Response: We agree that this type of analysis would be interesting, but because we only have those additional phenotypes measured in the AIS cases and not in the controls, we are unable to perform Mendelian randomization analysis.

Because you didn't include any co-variables in your Logistic model (not all of them have these data), the association should be interpreted carefully, and the limitation should be presented.

Response: We have added the following to the discussion: "We do not have these variables measured in controls. In future studies, incorporation of these variables as covariates or performing Mendelian randomization analysis might allow us to determine the causal relationships between these traits, each of which correlate with genotype at rs13107325. Currently, it is unknown whether genotype of this SNP affects scoliosis risk directly, or indirectly through its effect on other traits."

I am wondering which dataset you were using to generate Supplementary Figure 2, there was a significant SNP at p value of $\sim 10^{-9}$, however, this SNP was presented neither in table 1 nor in figure 1. What did the red line mean? if you combined some datasets, how about the second significant SNP in the figure 1?

Response: The dataset used in supplementary figure 2 was our exome cases vs. ExAC. We are now removing all ExAC analysis as per your suggestion so we are removing that figure from the supplementary information.

Not all of the cases were severe idiopathic scoliosis (Cobb angle $>40^\circ$), in your exome-wide study, the range was 10-180, in your replication study, the range was 20-104.

Response: We apologize for a typographical error, as the range of the exome-wide study was 20-180. We have added the words "moderate to severe" to the description of the replication cohort to indicate (as the range describes) that not all patients had "severe" AIS as defined as Cobb angles $>40^\circ$. Our discovery cohort was made up of 90% individuals with Cobb angles $>40^\circ$, so we stand by our assertion that the discovery cohort is a "severe" AIS cohort. We can, if need be, remove all reference to severity, but we believe it to be an important point in describing our cohorts as various previous studies have used idiopathic patients with little family history and curves in the 10-20 range.

In the Methods, "Cases were included only if Cobb angle $>20^\circ$ ", but in the Results, the range was 10-180.

Response: We have fixed the range to be 20-180. This was a typo.

"Ancestry was verified for all AIS cases and in-house controls using EIGENSTRAT", please show the results in supplementary figures.

Response: We have now included a supplementary figure showing the PCA plot for cases, controls and Hapmap reference populations.

"A subsample was checked for consistency (N=100) and there was complete concordance between genotyping methods", please show the results in supplementary tables.

Response: We have now included a supplementary table with the double genotyped individuals and their genotypes.

In discussion, this SNP is not a rare variant, it is common. "We identified a highly significant association between a moderately rare coding variant..."

Response: We have removed the description of the variant as rare.

Please refer figure 3 in the paragraph "In Vitro Functional Testing".

In line 126, it should be figure 2, right?

Response: We have added references to the figures.

Reviewers' comments:

Reviewer #2 (Remarks to the Author):

The authors have made edits and provided additional data further substantiating their findings. These changes and additions have solved remaining issues.

One tiny detail to edit before publication is to change "...locomotor defects during adolescence" for "locomotor defects during juvenile stages" as this statement refers to zebrafish and not patients.

Reviewer #4 (Remarks to the Author):

The manuscript is much more straightforward after 2 rounds of revision.

The authors failed to perform MR analysis to look at the causality effect of other traits and scoliosis, they had a discussion to this.

If the "severe" AIS was defined by Cobb angles >40deg, I would suggest the authors stick to the fact that, not all of the cases in the discovery were "severe" AIS. or the author could perform a subgroup analysis with only "severe" AIS as cases in discovery, to see if the results remain the same? Just like the authors declaimed that, this SNP was not found by other studies, it might be because this discovery cohort was "severe" AIS.

In line 280, "Given these sample sizes, this study had >80% power", What did you use to calculate the power? how you calculate the power?

Please do a pie plot for the MAF distribution for your discovery cohort, and presented as an supple fig.

I am a little curious about the 100% concordance, the genotypes of this SNP was extracted from Affy 6.0, and compared with the newly genotyped by KASPAR. Did you compare other SNPs, what were the concordance rate?

In line 111, the P value in combined sample was no longer $\sim 10^{-25}$, as you removed the ExAC in the combined analysis.

Responses to Reviewers' comments:

Reviewer #2 (Remarks to the Author):

The authors have made edits and provided additional data further substantiating their findings. These changes and additions have solved remaining issues.

One tiny detail to edit before publication is to change ".locomotor defects during adolescence" for "locomotor defects during juvenile stages" as this statement refers to zebrafish and not patients.

Response:

We agree that juvenile stages is a more appropriate description and have changed the wording in the text.

Reviewer #4 (Remarks to the Author):

The manuscript is much more straightforward after 2 rounds of revision.

The authors failed to perform MR analysis to look at the causality effect of other traits and scoliosis, they had a discussion to this.

If the "severe" AIS was defined by Cobb angles >40deg, I would suggest the authors stick to the fact that, not all of the cases in the discovery were "severe" AIS. or the author could perform a subgroup analysis with only "severe" AIS as cases in discovery, to see if the results remain the same? Just like the authors declaimed that, this SNP was not found by other studies, it might be because this discovery cohort was "severe" AIS.

Response:

Additionally, the association of rs13107325 with AIS remained when only severe AIS patients were used in the analysis (N=409) ($P = 7.7 \times 10^{-8}$; OR = 2.08, CI=1.58-2.73).

In line 280, "Given these sample sizes, this study had >80% power", What did you use to calculate the power? how you calculate the power?

Response:

We have added to the methods section a description of how the power analyses were performed: "We performed power calculations using the genetic power calculator available here:

<http://zzz.bwh.harvard.edu/gpc/>."

Please do a pie plot for the MAF distribution for your discovery cohort, and presented as an supple fig.

Response:

We have added a pie graph of the MAFs of SNPs analyzed in the discovery cohort as a supplementary figure.

I am a little curious about the 100% concordance, the genotypes of this SNP was extracted from Affy 6.0, and compared with the newly genotyped by KASPAR. Did you compare other SNPs, what were the concordance rate?

Response: We did not genotype other SNPs using KASPAR. We have added that we performed KASPAR genotyping for samples genotyped by either Affy 6.0 or exome sequencing and both

have a concordance rate of 100%.

In line 111, the P value in combined sample was no longer $\sim 10^{-25}$, as you removed the ExAC in the combined analysis.

Response: Thank you for catching this error. We have replaced the p-value with the correct p-value now.

REVIEWERS' COMMENTS:

Reviewer #4 (Remarks to the Author):

The authors explained all my concerns.

Reviewer #1 (Remarks to the Author on the B version of the manuscript):

The authors have addressed well my remaining points of concern and I have no further comments.
Please receive my apologies for the delay in responding to the journal.

REVIEWERS' COMMENTS:

Reviewer #4 (Remarks to the Author):

The authors explained all my concerns.

Response: We thank the reviewer for their many useful comments in previous revisions.

Reviewer #1 (Remarks to the Author on the B version of the manuscript):

The authors have addressed well my remaining points of concern and I have no further comments. Please receive my apologies for the delay in responding to the journal.

Response: We thank the reviewer for their many useful comments in previous revisions.